# Neurogenetic phenotypes of learning-dependent plasticity for improved perceptual decisions
Liz Yuanxi Lee [1], Joseph J. Ziminski[1,2], Polytimi Frangou [1,3,4], Vasilis M. Karlaftis [1], Yezhou Wang [5], Boris Bernhardt [5], Varun Warrier [1,6], Richard A. I. Bethlehem [1] & Zoe Kourtzi [1,7]✉

Genetics and experience are known to mold our cognitive development. Yet, the interactions between genetics and brain mechanisms that support learning and flexible behavior in the adult human brain remain largely unknown. Here, we test the link between brain-wide gene expression and macroscopic neuroimaging phenotypes of brain plasticity that support our ability to improve perceptual decisions with training. We demonstrate that gene expression links to learning-dependent changes in spatial variations of cortical microstructure and functional connectivity in visual and fronto-parietal networks that are known to be involved in perceptual decisions. Further, we show that brain stimulation in visual cortex during training boosts learning and alters functional connections, rather than microstructure organization, within and between these networks. Our results reveal neurogenetic phenotypes of plasticity in perceptual decision networks, providing insights into the interplay of genetic expression and macroscopic mechanisms of structural and functional plasticity for learning and flexible behavior.

Successfully interacting in our complex environments requires selecting information that is key for identifying targets in cluttered scenes; for example, spotting a prey camouflaged in natural scenes or identifying a friend in the crowd. It is known that experience and training enhance these skills by altering the brain's structural organization (for review[1]) and functional connections (for reviews[2,3]). Previous studies have shown that training in a range of tasks (e.g. perceptual, motor tasks) results in changes in functional connectivity[1]. Further, training has been shown to promote myelination in the adult brain, the process of insulating neural axons to enhance neurotransmission (for reviews[4,5]), that is known to play an active role in learning and brain plasticity[5,6].

Complementary to these brain plasticity mechanisms, there is evidence for genetic factors contributing to learning. For example, brain-derived neurotrophic factor (*BDNF*) and the extracellular signaling-related kinases (*ERKs*) are found to promote both developmental and learning-related plasticity[7]. Further, knockout of *Fmr1* that produces fragile *X[fra(X)]* syndrome alters behavioral performance in a complex discrimination task for mice[8]. Yet, the interactions between genetic and brain plasticity mechanisms that contribute to our ability to improve perceptual decisions with training—a skill known as perceptual learning (for review[9])—remain largely unknown.

Here, we test the link between brain-wide gene expression (Fig. 1A) and learning-dependent changes in microstructure and functional connectivity (Fig. 1B) due to training on a visual discrimination task that involves identifying patterns embedded in noise (Signal-in-noise, SN task; Fig. 2A). Using the Allen Human Brain Atlas (AHBA)[10], a comprehensive brain-wide gene expression atlas, we test whether spatial variations at the transcriptomic level are associated with macroscopic neuroimaging signatures of brain plasticity. We employ a multimodal brain imaging approach to investigate microstructural (i.e. myelination) and functional (i.e. functional connectivity) plasticity mechanisms. We use: a) quantitative MRI multi-parameter mapping (MPM[11]) to measure a myelin-sensitive magnetic resonance imaging marker (i.e. magnetization transfer saturation (MT) b) resting state fMRI (rs-fMRI) to measure functional connectivity. We employ gradient analysis[12] of MT and functional connectivity signals to characterize principal gradients of cortical microstructure (i.e. grey matter myelin) organization and functional connectivity that capture overarching spatial trends extending beyond local processing (i.e. local functional connectivity, regional grey and white matter volume). Previous work has demonstrated that gradient analysis offers a spatial framework for organizing multiple large-scale networks. Further, it characterizes a

¹Department of Psychology, University of Cambridge, Cambridge, UK. ²Sainsbury Wellcome Centre, University College London, London, UK. ³Centre for Integrative Neuroimaging, FMRIB, Nuffield Department of Clinical Neurosciences, University of Oxford, Oxford, UK. ⁴MRC Brain Network Dynamics Unit, Nuffield Department of Clinical Neurosciences, University of Oxford, Oxford, UK. ⁵McConnell Brain Imaging Centre, Montreal Neurological Institute and Hospital, McGill University, Montreal, QC, Canada. ⁶Department of Psychiatry, University of Cambridge, Cambridge, UK. ⁷Dept of Psychology, Justus-Liebig University, Giessen, Germany. ✉e-mail: zk240@cam.ac.uk

## A. Gene Regional Expression

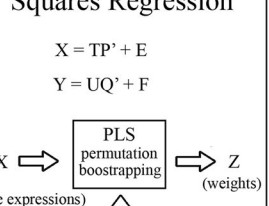

## C. Partial Least Squares Regression

$$X = TP' + E$$
$$Y = UQ' + F$$

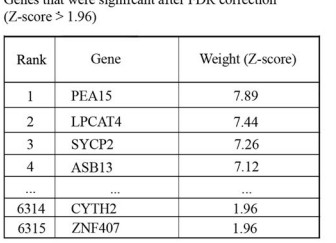

(gene expressions) X ⇒ PLS permutation boostrapping ⇒ Z (weights)

Y (cortical changes)

## D. List of Genes from PLS

Genes that were signifcant after FDR correction (Z-score > 1.96)

| Rank | Gene | Weight (Z-score) |
|---|---|---|
| 1 | PEA15 | 7.89 |
| 2 | LPCAT4 | 7.44 |
| 3 | SYCP2 | 7.26 |
| 4 | ASB13 | 7.12 |
| ... | ... | ... |
| 6314 | CYTH2 | 1.96 |
| 6315 | ZNF407 | 1.96 |

## E. Enrichment Test

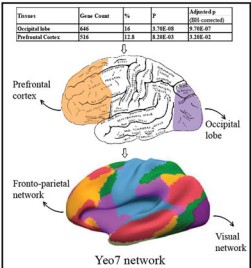

## B. Cortical Microstructural and Functional Connectivity Gradient

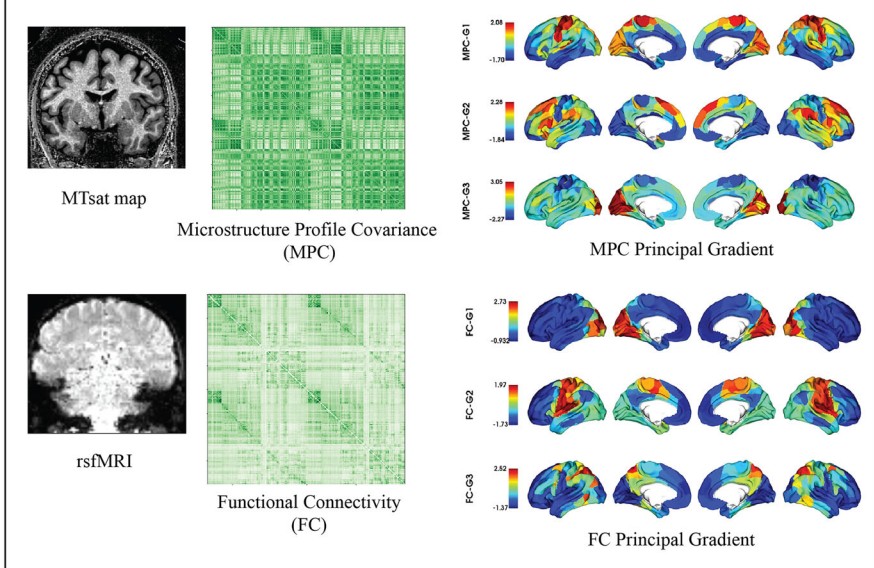

MTsat map

Microstructure Profile Covariance (MPC)

MPC Principal Gradient

rsfMRI

Functional Connectivity (FC)

FC Principal Gradient

## F. Network Dispersion

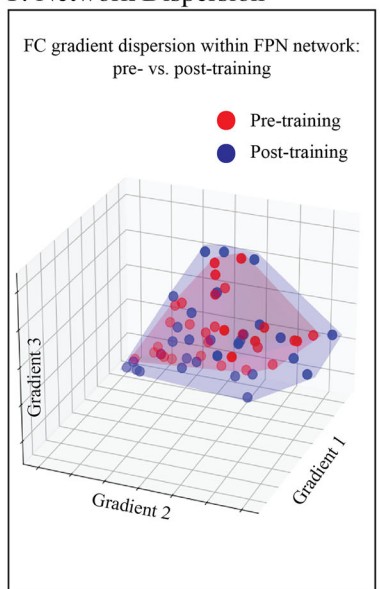

FC gradient dispersion within FPN network: pre- vs. post-training

● Pre-training
● Post-training

**Fig. 1 | Schematic overview linking genetic expression and gradient analyses.**
**A** The median gene expression profiles for 16651 genes were calculated across 200 cortical nodes using Schäefer-200 parcellation. Due to the limited availability of data, we focused on left hemisphere, where gene expression data are available for all six donors. **B** Microstructural (derived from MT maps) and functional (derived from rs-fMRI) gradients were calculated across the same 200 nodes (for Schäefer 300, 400 parcellations, see Fig. S2) as the gene expression analysis. Color maps represent gradient values. **C** PLS regression analysis was performed with gene expression profiles as predictors and cortical changes as response variables for $n = 10{,}000$ permutations. **D** PLS assigns weights to each gene indicating its contribution to the overall model for each component. Bootstrapped standard errors were derived and the gene weights were Z-transformed and corrected for multiple comparison using FDR inverse quantile transform correction to account for winners curse. **E** Genes that were significant after FDR correction (z-score > 1.96) were enriched for tissue types. Genes in PLS component 1 showed significant enrichment; these genes (Table S1 for top ranked 200 genes) are preferentially expressed within occipital and prefrontal brain regions, corresponding to visual and fronto-parietal networks (Yeo7 network). **F** FC within-network dispersion in the fronto-parietal network (FPN). Red dots, blue dots and the shaded areas represent the dispersion of principal gradients for pre- and post-training sessions, respectively. The shaded area for the post-training session (blue) is larger than that for the pre-training session (red), indicating that the nodes representing functional connectivity similarity within the FPN network are more spread-out after training (i.e. higher network segregation). For 3D illustration see Fig. S1.

spectrum of functional activity, ranging from unimodal to heteromodal regions, as demonstrated by functional meta-analyses[12–14] and micro-structural gradients analyses[15,16]. Finally, gradient analysis allows us to link and compare microstructure and functional plasticity in the same information space by extracting markers of structural and functional brain plasticity based on the similarity spatial gradients within and across cortical networks. Combining these gradient analyses of multimodal brain imaging data with genetic analyses, we ask whether gene expression in the adult human brain links to learning-dependent changes in cortical microstructure and functional connectivity that relate to learning (i.e. improved performance in the SN task).

Our results demonstrate an interplay of brain-wide genetic expression with functional and microstructural plasticity mechanisms for learning in the adult brain. We show that genetic expression in occipital and prefrontal regions is associated with learning-dependent changes in microstructure organization and functional connectivity. In particular, higher micro-structural coherence between visual and fronto-parietal networks, micro-structural segregation within the visual network and functional segregation

of regions within the fronto-parietal network relate to faster learning during multi-session training. Further, we employ brain stimulation during brain imaging to directly test the link between these neurogenetic phenotypes of plasticity and improved perceptual decisions due to training. Our results demonstrate that anodal tDCS in visual cortex during training within a single session boosts learning and alters functional connectivity within and between visual and fronto-parietal networks rather than microstructural organization. Our results provide evidence for a direct link between gene expression in perceptual decision networks, functional plasticity at early stages of learning, and both functional and microstructural plasticity at later stages of learning (i.e. following longer-term training) in the adult human brain.

## Results

### Behavioral performance

We trained participants in a signal-in-noise (SN) task that involved iden-tifying radial vs. concentric Glass patterns (Fig. 2A) embedded in noise[17]. Participants completed three behavioral sessions during MRI scanning

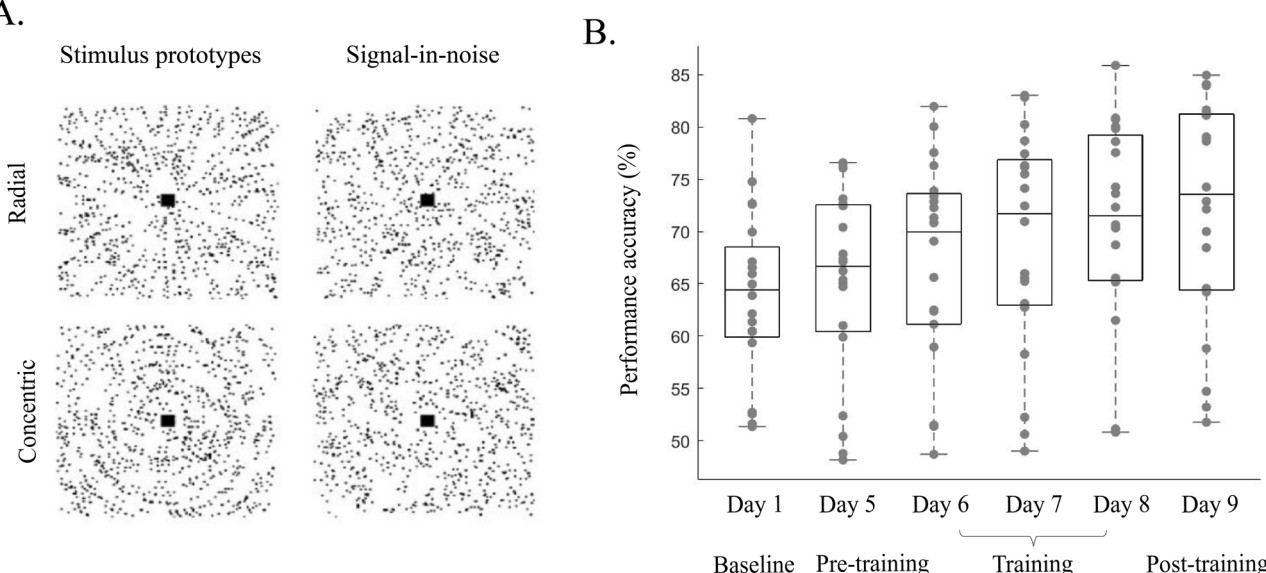

**Fig. 2 | Experimental design and stimuli. A** Example Radial and concentric Glass patterns are shown with inverted contrast for illustration purposes. Left: Prototype stimuli: 100% signal, spiral angle 0˚ for radial and 90˚ for concentric are shown for illustration purposes only. Right: Stimuli used in the study: 25% signal, spiral angle 0˚ for radial and 90˚ for concentric. **B** Behavioral accuracy (%) across participants for each session ($n = 22$). Grey dots indicate individual participant data. Box plots indicate median (black line), 25th and 75th percentiles (box bottom, top edges respectively), data range (whiskers).

without feedback (day 1: baseline, day 5: pre-training, day 9: post-training) and three consecutive task training behavioral sessions with feedback (day 6, day 7, day 8). Training improved participant performance in the task (Fig. 2B.; one-way repeated measures ANOVA across sessions, Greenhouse-Geisser corrected, $F(2, 38) = 35.49$, $p < 0.001$); that is, performance accuracy increased in the post- vs. pre-training sessions (post-hoc comparisons, $p < 0.001$, Bonferroni corrected). This improvement was specific to training; that is, there were no significant differences in performance between baseline vs. pre-training sessions ($p = 0.47$).

## Genetic signatures of learning-dependent plasticity in occipital and prefrontal regions

We tested for genes that contribute to learning-dependent brain plasticity following training on the SN task; that is, changes (post- vs. pre-training) in microstructural and functional cortical organization, as measured by MT and rs-fMRI, respectively. To capture microstructure and functional connectivity (FC), we used gradient analyses to map macroscale brain features to low dimensional spatial representations[12]. We followed recent work that implemented gradient analysis to spatial topography of MT profiles[18] and hierarchical functional organization of the cortex[19]. In brief, we created cortical surface models from T1-weighted MRI scans and aligned the corresponding MT maps to these surfaces. We then generated equivolumetric intracortical surfaces and sampled MT intensities along vertices perpendicular to the cortical mantle to construct intracortical MT profiles (Methods - Multi-session training study: Data analysis)[18]. We next computed changes (i.e. post- vs pre- training) in microstructure profile covariance (MPC) and FC principal gradients (Fig. 1B).

To test the link between genetic expression and learning-dependent changes in microstructural and functional plasticity, we used partial least squares regression (PLS) regression (Fig. 1C; Methods - Multi-session training study: Gene expression analysis) to identify independent components (i.e. linear combinations of gene expressions) that relate to learning-dependent changes in MPC and FC principal gradients. In particular, we used the Allen Human Brain Atlas[10] to test whether gene expression in each node (Fig. 1A) relates to changes (i.e. post- vs pre-training) in the spatial variation of microstructure (i.e. MPC principal gradients: G1-G3) and functional connectivity (i.e. FC principal gradients: G1-G3) profiles (Fig. 1B). We found that the first three PLS components explained

significant variance in learning-dependent microstructure and functional changes (10,000 permutations test; PLS1: $p = 0.036$, PLS2: $p = 0.0081$, PLS3: $p = 0.038$). Specifically, these components accounted for 5.32% (PLS1), 5.68% (PLS2), 3.87% (PLS3) of the variance. PLS weights for each gene were z-transformed (based on bootstrapping, $n = 1000$) and FDR-adjusted (Fig. 1D). We then conducted a gene enrichment test (tissue-type specific, GNF U133A quartile 79 tissue types[20]) to test for tissue-specific gene expression and regulation (Fig. 1E). Genes in PLS components 1 ($n = 4225$) and 2 ($n = 71$) that passed FDR correction ($p < 0.05$[21]) were included in the enrichment analysis (no genes in PLS component 3 passed FDR correction; Table S1 for top 200 significant genes after FDR correction). This analysis identified significant enrichment for genes in PLS component 1 that are preferentially expressed within occipital ($p < 0.001$, Benjamin-Hochberg corrected) and prefrontal ($p = 0.032$, Benjamin-Hochberg corrected) lobes among brain tissues (see Table S2 for all significant enriched tissue types; no significant enrichment results were found for PLS component 2). To account for spatial auto-correlations in gene expression[22], we generated 1000 null brain models for gradients using the brainspace toolbox[23] and conducted the PLSr analysis with a) raw gene expression data (Table S3), b) denoised gene expression data, i.e. with 1st PCA component regressed out from the raw gene expression to account for overexpression in the occipital lobe[24] (Table S4). This analysis showed that gene expression from the same regions (i.e. occipital, prefrontal) was significantly enriched. These results suggest that gene expression preferentially in the occipital and prefrontal cortex is associated with changes in structural organization and functional connectivity due to training on a perceptual discrimination task.

## Learning-dependent changes in occipito-frontal brain networks

We next asked whether learning-dependent changes in microstructure and functional connectivity that were shown to link to gene expression in occipital and prefrontal cortex relate to learning in the SN task. We extracted MPC and FC gradients for the visual network (VN) including occipital regions and the frontoparietal network (FPN) including prefrontal regions based on a seven functional brain networks atlas[25] (Fig. 1E). Recent work has employed gradient analysis to spatial topography of age-related changes in MT profiles[15] and hierarchical functional organization of the cortex[19]. We then estimated within- and between-network dispersion for MPC and FC principal gradients, a metric developed to quantify the variability in

**Fig. 3 | Learning-dependent changes in occipito-frontal brain networks. A** Correlation between VN-FPN between-network MPC dispersion changes (post- vs. pre-training) and learning rate ($p = 0.017$, $R = -0.59$, CI [$-0.84$, $-0.13$]). **B** Correlation between FPN within-network dispersion changes (post- vs. pre-training) of principal FC gradients and learning rate ($r = 0.668$, $p = 0.0024$, CI [0.29, 0.87]). **C, D**. Multiple regression analyses showing that changes in MPC dispersion explain significantly variance in behavioral improvement with changes in (**C**) VN within-network MPC dispersion and (**D**) VN-FPN between-network MPC dispersion as significant predictors of behavioral improvement.

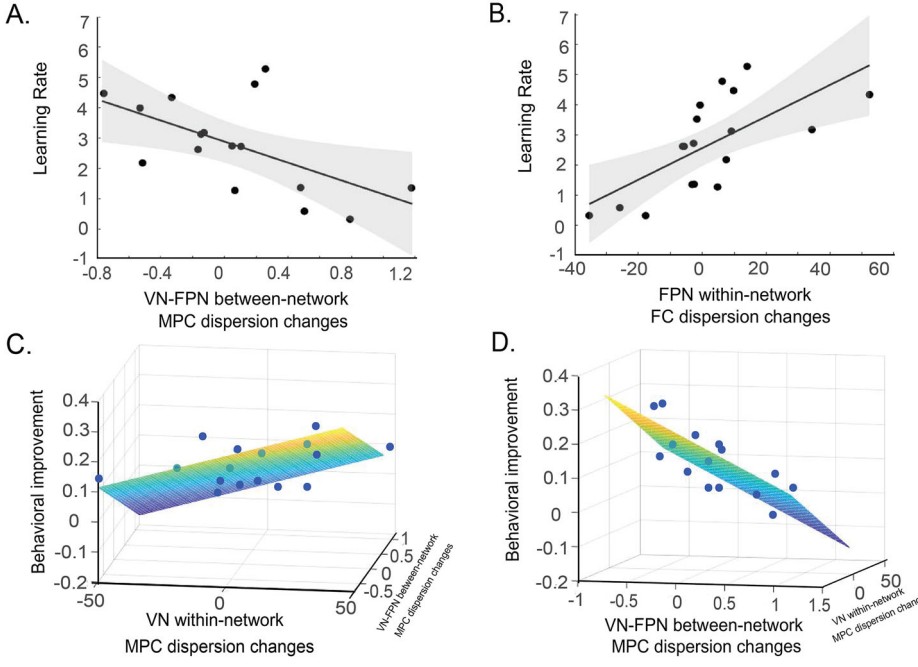

structural complexity and connectivity patterns across different brain regions. We defined the dispersion space by the values along the first three gradients (Fig. 1F, Fig. S1). Within network dispersion was quantified as sum squared Euclidean distance of network nodes to the network centroid at individual participant level. Between network dispersion was calculated as the Euclidean distance between network centroids. Dispersion indicates microstructural and functional network segregation[19] (Fig. 1F, Fig. S1; Methods—multi-session training study: Data analysis). That is, higher dispersion indicates higher segregation, while lower dispersion indicates higher coherence within or between networks.

We tested whether changes (post- vs. pre-training) in MPC and FC dispersion account for variance in behavioral performance (i.e. learning rate) using multiple regression analyses. Our results showed that changes in MPC dispersion accounted significantly for variance in learning rate ($F(3, 12) = 4.42$, $p = 0.026$, $R^2 = 0.53$, $R^2_{Adjusted} = 0.41$; permutation test, $n = 1000$, $p = 0.028$; Fig. 3A). In particular, changes in VN-FPN between-network MPC dispersion (Beta = $-0.73$, $t(15) = -3.21$, $p = 0.007$) were significant predictors of learning rate. However, changes in VN (Beta = 0.51, $t(15) = 2.02$, $p = 0.067$) and FPN (Beta = 0.38, $t(15) = 1.68$, $p = 0.118$) within-network MPC dispersion were not significant predictors of learning rate. Further, we found that changes in FC dispersion accounted significantly for variance in learning rate ($F(3, 14) = 7.10$, $p = 0.004$, $R^2 = 0.60$, $R^2_{Adjusted} = 0.52$; Fig. 3B). In particular, changes in FPN within-network FC dispersion (Beta = 1.04, $t(17) = 4.16$, $p = 0.001$) were significant predictors of learning rate (permutation test, $n = 1000$, $p = 0.001$). However, changes in VN within-network FC dispersion (Beta = $-0.05$, $t(17) = -0.25$, $p = 0.806$) and VN-FPN between-network FC dispersion (Beta = 0.56, $t(17) = 2.11$, $p = 0.053$) were not significant predictors of learning rate. Figure 3B shows that increased dispersion in FPN functional connectivity relates to faster learning rate, suggesting that higher functional segregation of regions within this network after training relates to learning. In contrast, decreased microstructure dispersion between VN and FPN (Fig. 3A) relates to faster learning rate, suggesting that higher coherence in structural organization between networks after training relates to learning.

To test whether these learning-dependent changes were specific to training, we conducted permutation analyses (1000 iterations) on regression models using random pairings between baseline and pre-training data for EV within-network, FPN within-network, and EV-FPN between-network dispersion. Only 46/1000 models for FC dispersion ($p = 0.954$) and 45/1000

models for MPC dispersion ($p = 0.955$) indicated a relationship between pre- vs. post-training differences and behavior. These findings support the specificity of learning-dependent changes, suggesting that baseline vs. pre-training differences are unlikely to contribute significantly to the learning effects we observed.

We next tested whether learning-dependent changes (before vs. after training) in MPC and FC dispersion accounted for variance in behavioral improvement (i.e. performance accuracy before vs. after training). Multiple regression analysis showed that changes in MPC dispersion ($F(3, 15) = 6.23$, $p = 0.009$, $R^2 = 0.609$, $R^2_{Adjusted} = 0.511$) rather than changes in FC dispersion ($F(3, 14) = 3.24$, $p = 0.054$) accounted significantly for variance in behavioral improvement. In particular, changes in VN within-network MPC dispersion (Beta = 0.648, $t(15) = 2.84$, $p = 0.015$) and VN-FPN between network MPC dispersion (Beta = $-0.831$, $t(15) = -4.06$, $p = 0.002$), rather than FPN (Beta = 0.26, $t(15) = 1.26$, $p = 0.23$) within-network dispersion, were significant predictors of behavioral improvement. Figure 3C shows that increased microstructure dispersion in VN relates to higher behavioral improvement, suggesting that higher segregation in structural organization across regions in the visual network after training relates to learning. In contrast, decreased microstructure dispersion between VN and FPN (Fig. 3D) relates to higher behavioral improvement, suggesting that higher coherence in structural organization across networks after training relates to learning. These results were specific to the training; that is, changes in MPC before training (pre-training vs. baseline) did not account significantly for variance in changes in performance accuracy after training ($F(3, 12) = 0.63$, $p = 0.61$; Supplementary Information: Changes in occipito-frontal brain networks before training (pre-training vs. baseline)).

Taken together, our results suggest that learning relates to increased microstructure coherence between visual and fronto-parietal networks, as indicated by significant relationships of learning-dependent changes in microstructure with learning rate and behavioral improvement (Fig. 3A, C). Interestingly, higher functional segregation of regions within the fronto-parietal network relates to faster learning (Fig. 3B), while higher structural segregation of regions within the visual network relates to higher improvement (Fig. 3D) in performance after training. These results suggest that training alters functional connectivity within the frontal cortex to accelerate learning, while microstructure organization within visual cortex to support improved performance following training. Further, higher

structural coherence between visual and frontal networks supports both faster learning and improved performance after training.

## Anodal tDCS during training and brain imaging

To directly test the link between microstructure and functional plasticity with behavior, we used anodal tDCS stimulation in visual cortex. Anodal tDCS has been previously shown to facilitate learning[26] and improve performance in visual discrimination tasks[27]. We have previously shown that participants trained during Anodal tDCS improved in the SN task compared to Sham and no-training control groups[26]. Here, we used tDCS in the scanner to test whether Anodal stimulation in visual cortex during training on the SN task alters microstructure and functional plasticity in visual and fronto-parietal networks. We trained two groups: one group received excitatory (Anodal) stimulation during training, while the other received sham stimulation in visual cortex) during training. We collected MPM and rs-fMRI data before and after stimulation within the same imaging session, extracted MPC and FC gradients from VN and FPN and compared within-network and between-network dispersion before and after stimulation with training.

**Anodal tDCS improves performance in signal-in-noise discrimination.** Participants who received anodal stimulation showed improved behavioral performance in the SN task compared to participants who received sham stimulation. In particular, a two-way repeated measures ANOVA on behavioral performance showed a significant Group (Anodal, Sham) x Block (Pre-, Post-stimulation) interaction ($F(1, 43) = 4.72$, $p = 0.035$; Fig. 4A), but no significant main effect of Block ($F(1, 43) = 1.32$, $p = 0.26$) nor significant main effect of Group ($F(1, 43) = 2.98$, $p = 0.091$). Post-hoc comparison showed that participants in the Anodal group ($p = 0.019$, Bonferroni corrected) improved significantly after training. In contrast, there was no significant improvement for the Sham group ($p = 0.49$). Further, we didn't observe significant differences in performance between groups before stimulation ($p = 0.31$, $t(43) = 1.02$), suggesting that our findings are unlikely to be simply due to variability across participants in task performance before training. Our results suggest that anodal stimulation during training boosts learning, providing an independent replication of our previous findings showing a facilitatory effect of anodal tDCS on behavioral improvement in visual discrimination tasks[26].

**Anodal tDCS alters microstructure and functional connectivity in occipito-frontal networks.** We next asked whether anodal tDCS during training alters microstructure and functional connectivity in occipito-frontal networks. To directly investigate the effect of this intervention during training, we normalized microstructural and functional dispersion for the Anodal to the Sham group data, as we didn't observe significant differences between groups before stimulation in (a) behavioral performance ($t(37) = 1.02$, $p = 0.31$), (b) MPC dispersion (VN within-network: $t(37) = 33.69$, $p = 0.49$; FPN within-network: $t(37) = 28.45$, $p = 0.68$; VN-FPN between-network MPC dispersion: $t(37) = 36.78$, $p = 0.63$), (c) FC dispersion (VN within-network: $t(34) = 29.13$, $p = 0.84$; FPN within-network: $t(34) = 26.74$, $p = 0.81$; VN-FPN between-network FC dispersion: $t(34) = 28.54$, $p = 0.58$,).

Our results showed that anodal tDCS during training altered significantly functional connectivity rather than microstructure. In particular, a two-way repeated measures ANOVA (Network: VN, FPN, VN-FPN between-network; Block: Pre-, Post-stimulation) for FC dispersion showed a significant Network × Block interaction ($F(2, 42) = 25.12$, $p < 0.001$). We did not observe a significant main effect for Network ($F(2, 42) = 0.512$, $p = 0.60$) nor Block ($F(1, 21) = 2.80$, $p = 0.12$). Post-hoc comparisons showed that VN within-network FC dispersion significantly decreased after training ($p = 0.006$, Bonferroni corrected; Fig. 4B), while FPN within-network FC dispersion significantly increased after training ($p < 0.001$, Bonferroni corrected; Fig. 4B). Further, VN-FPN between-network FC dispersion decreased after training (p = 0.023, Bonferroni corrected;

Fig. 4B). Finally, these results were specific to FC dispersion; that is, we did not observe a significant Network x Block interaction for MPC dispersion ($F(2, 40) = 2.23$, $p = 0.12$). These results suggest that anodal tDCS in visual cortex during training boosts learning and functional connectivity in occipito-frontal networks; that is, increasing functional segregation within frontal cortex (Fig. 4B), while increasing coherence across occipital regions (Fig. 4B) and between occipital and fronto-parietal networks (Fig. 4B). Note that these changes were measured before vs. after the tDCS intervention when there was no tDCS stimulation or performance feedback provided to the participants. tDCS may enhance attention during the intervention. However, following the intervention task performance was improved suggesting that the task was easier and therefore less attentionally demanding in comparison to the Sham group, where the task remained demanding and may have required more attentional resources.

## Discussion

We provide evidence for neurogenetic phenotypes of plasticity, linking brain-wide gene expression, macroscopic connectome organization and functional brain plasticity for improved perceptual decisions. We demonstrate that genetic expression in occipito-frontal regions links to learning-dependent changes in spatial variations of cortical microstructure and functional connectivity profiles that relate to perceptual learning. In particular, higher structural coherence between visual and fronto-parietal networks supports both faster learning and improved performance after training. Further, higher functional segregation of regions within the fronto-parietal network relates to faster learning, while higher microstructural segregation within the visual network relates to higher improvement after training. Next, we demonstrate that anodal tDCS stimulation in the visual cortex during training boosts learning and alters functional connectivity rather than microstructure organization. Our findings advance our understanding of the interplay of genetic expression with functional and microstructural plasticity mechanisms for learning in the adult brain in the following respects.

First, we identify sets of genes that are associated with learning-dependent changes in cortical microstructure organization and functional connectivity due to training in a visual discrimination task. Most previous studies have focused on the associations between single genes or a small numbers of genes and cognitive functions. Here, using AHBA and enrichment tests, we link wide-brain genetic expression (i.e. across gene sets) with learning-dependent structural and functional plasticity in brain networks. Our analyses revealed genes that are known to be associated with: a) cognitive flexibility and brain plasticity (e.g. *APOE*[28], b) implicit learning (e.g. *OXTR*[29], (c) learning and memory (e.g. *Arc*[30], *FOXP2*[31], *ERK/CREB* pathway[32]), d) prefrontal function (e.g. *COMT*[83]), and striatal neuroplasticity (e.g. *FOXP2*[31]). Although *BDNF* has been previously implicated in learning and brain plasticity[7], our results did not show a significant link of *BDNF* to learning-dependent changes in cortical microstructure and functional connectivity. This is in agreement with previous work implicating *BDNF* in associative (e.g. spatial learning and memory, paired associated learning)[34], rather the perceptual learning. Interestingly, the enrichment test revealed that genetic expression preferentially in occipital and prefrontal regions relates to learning-dependent changes in microstructure and functional connectivity, consistent with the role of these regions in perceptual decision-making, for reviews[35,36].

Our results offer potential insights into the transcriptomic bases of a macroscale brain phenotype that relates to perceptual learning; however, it is important to consider potential limitations associated with extracting gene expression patterns using the AHBA dataset. First, AHBA data are extracted from only six healthy donor brains, with most samples taken from the left hemisphere[10]. Further, the brain tissue samples used for RNA sequencing in the AHBA were not homogeneously distributed across the cortex[37]; resulting in estimates of regional expression based on different numbers of experimental measurements in each of the 200 regions. Importantly, gene expression is influenced by factors such as sex, age, genetics, and environment[38]. Future work is needed, considering a broader range of

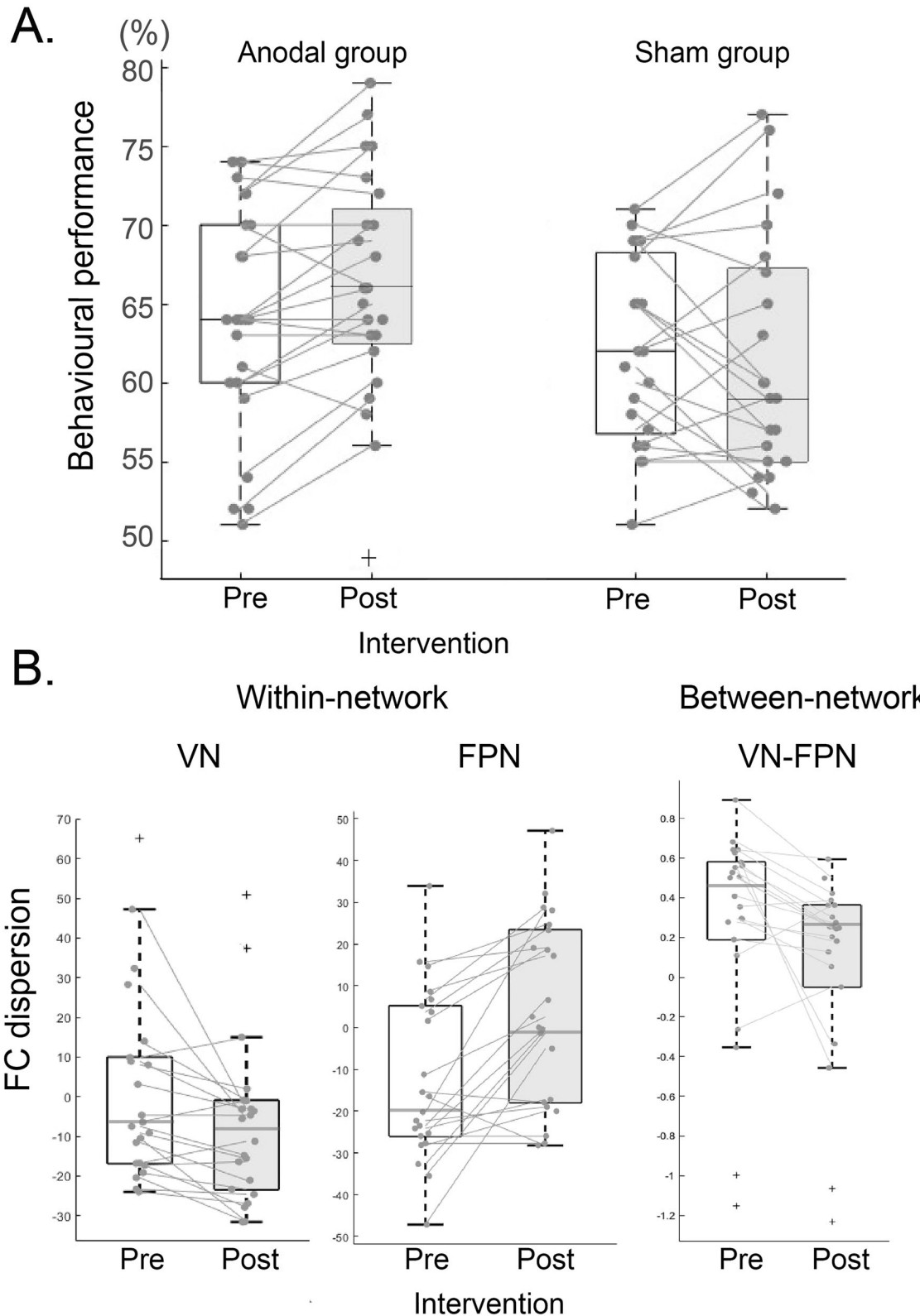

**Fig. 4 | Intervention-dependent changes in behavior and occipito-frontal brain networks. A** Behavioral performance (percent) across participants per group (Anodal, Sham) and Block (Pre-, Post-intervention). **B** Functional connectivity dispersion before vs. after intervention (Anodal tDCS stimulation during training, normalized to Sham stimulation) for a visual within-network, b fronto-parietal within-network c VN-FPN between-network. Grey dots indicate individual participant data. Box plots indicate median (black line), 25th and 75th percentiles (box bottom, top edges respectively), data range (whiskers). Outliers are indicated by '+'.

individual variability and using methods for estimating gene expression at the individual level (e.g. transcriptome-wide association study; TWAS[39]), to identify relationships between gene expression and specific human traits. TWAS studies leverage large population-specific datasets to enhance statistical power. Although there are no existing large-scale GWAS data on perceptual learning and our study sample is not large enough for TWAS analysis, exploring gene-trait associations remains an interesting direction for future research.

Second, we employ a gradient analysis approach that allows us to link microstructure and functional plasticity at the same information; that is, capturing dispersion in spatial gradients within and across cortical networks as markers of structural and functional brain plasticity[12]. Most studies using gradient analyses have focused on developmental, aging[18,19,40] and disease-related[41,42] changes in microstructure and functional gradients. Extending beyond this previous work, we adopt gradient analyses to test whether training alters structural and functional cortical organization to support our ability for improved perceptual decisions, revealing macroscopic plasticity markers of plasticity in the adult brain. Interestingly, higher functional segregation of regions within the fronto-parietal network relates to faster learning, while higher microstructural segregation within the visual network relates to higher improvement after training. These findings suggest that learning alters functional compared to structural organization at different temporal and spatial scales within visual and fronto-parietal networks. That is, fronto-parietal plasticity supports learning during training, consistent with the role of this network as a flexible hub for cognitive control[43,44], while visual plasticity may support learning-dependent changes in sensory processing after training.

Most human imaging studies examining structural plasticity have employed diffusion tensor imaging (DTI) to investigate structural changes due to extensive training on visual and motor tasks (for a review[1]). Extending beyond this work, we combine quantitative MRI and gradient analysis to quantify spatial microstructural profiles that are sensitive to myeloarchitecture across the cortical surface[45]. Recent work has shown that multiparametric maps are highly reliable and—despite measuring myelin indirectly correlate closely with histological measures of myelin content[45]. Further, quantitative MRI (i.e. MPM) enables measurements of myelin in grey matter that has been linked to micro-circuit functions related to learning[5], rather than only white matter. Using this quantitative imaging approach, we have previously shown that training on the SN task alters subcortical myelination and functional connectivity to visual cortex. Here, we extend this work to cortical networks showing that higher cortical surface microstructural coherence between visual and fronto-parietal networks supports both faster learning and improved performance after training. Further, higher microstructural segregation within the visual network relates to higher behavioral improvement, suggesting that learning-dependent changes in visual network microstructure may support regional specialization for improved perceptual decisions after training on a visual discrimination task.

Further, previous neuroimaging studies have implicated visual networks in sensory processing, while fronto-parietal networks in information accumulation for perceptual decisions[46]. Our findings advance our understanding of learning-dependent functional plasticity mechanisms within and across visual and fronto-parietal networks for perceptual decisions. In particular, higher functional segregation of regions within the fronto-parietal network relates to faster learning, suggesting that learning-dependent changes in functional connectivity may support regional specialization within the fronto-parietal network for improved perceptual decisions.

Third, we demonstrate that anodal tDCS stimulation in the visual cortex during training alters functional connectivity and improves task performance rather than microstructure organization. Previous studies have shown that anodal tDCS alters functional connectivity during training in a range of tasks: e.g. associative learning[47], visual selective attention[48], and object identification[49]. Further, transcranial electric stimulation has been shown to facilitate visual perceptual learning in the context of perceptual tasks (for reviews[50,51]).

Comparing the results between longer-term (multi-session) training on the SN task and short-term brain stimulation during (single session) training provides insights into the brain plasticity mechanisms across stages of learning. First, we show that functional segregation within the fronto-parietal network: (a) relates to faster learning across training sessions, (b) is enhanced by anodal tDCS at early learning stages (i.e. single session training), suggesting that functional specialization within the fronto-parietal network may accelerate learning, consistent with the role of these regions in flexible control[25,52] and evidence accumulation for perceptual decisions. Second, we show that anodal tDCS at early learning stages alters functional connectivity but not microstructure organization within and between visual and fronto-parietal networks. In contrast, learning-dependent changes in microstructure but not functional connectivity within the visual network and between networks relates to improved task performance after longer-term training. These results suggest flexible functional plasticity at early stages of learning, while slower microstructural plasticity may support learning consolidation after longer-term training. Future work adopting brain stimulation during longer-term training is needed to further interrogate the interplay of microstructural and functional brain plasticity mechanisms across stages of learning.

In sum, our findings demonstrate that genetic expression in occipito-frontal regions underlies learning-dependent changes in microstructure organization and functional connectivity that relate to learning for improved perceptual decisions. Our results propose a tight interplay between genetic expression, microstructure organization and functional connections, revealing neurogenetic phenotypes of plasticity that support learning for perceptual decisions in the adult brain.

## Methods
### Overview
We conducted two studies: (a) multi-session training study, (b) intervention study. For both studies, participants were trained to discriminate radial vs. concentric Glass patterns embedded in noise. The multi-session training study investigated the link between gene expression, and learning-dependent functional and microstructural plasticity in perceptual decision networks. In this study, participants underwent multiple training sessions. MRI and behavioral test data were collected at baseline, pre- and post-training sessions. In the intervention study, we used anodal tDCS stimulation in visual cortex during scanning to compare learning-dependent network changes between the stimulation (Anodal tDCS during training) and the no-stimulation (Sham tDCS during training) groups. This allowed us to test directly the link between microstructure and functional plasticity with behavioral improvement. For both studies, we collected resting-fMRI and MPM data before and after training. We generated the microstructural profile covariance (MPC) matrix for cortical myelination densities and functional connectivity (FC) similarity matrix for resting-state data. We then used MPC and FC to build microstructural and functional gradients and calculate network dispersion that reflects the network segregation and coherence.

### Participants
Twenty-two healthy volunteers (23.5 ± 4.2 years) participated in the multi-session training study[17]. Forty-five healthy volunteers (27 female; mean age 22.9 ± 3.3 years) participated in two intervention groups, twenty-four in the stimulation group (Anodal) and twenty-one in the no stimulation group (Sham). Sample size was determined by power calculations following our previous work[26] showing a polarity-specific stimulation effect of $\eta_p^2 = 0.133$ at 95% power for N = 14 per group. All participants were right-handed, had normal or corrected-to-normal vision, did not receive any prescription medication, were naïve to the aim of the study, gave written informed consent and received payment for their participation. The study was approved by the University of Cambridge Ethics Committee [PRE.2017.057]. All ethical regulations relevant to human research participants were followed.

## Multi-session training study: Experimental Design and Procedures

As described in our previous study[17], participants were trained to discriminate radial vs. concentric Glass patterns embedded in noise (signal-in-noise task, SN; Fig. 2A, see also Supplementary Information: *Stimuli*). Each volunteer participated in six sessions: three brain-imaging sessions including testing on the SN task without feedback (day 1: baseline, day 5: pre-training, day 9: post-training) and three consecutive task training behavioral sessions with feedback (day 6, day 7, day 8). For more detailed experiment design see Supplementary Information: *Multi-session training study: Behavioral session design*. We recorded participants' behavioral performance and computed learning rate by fitting individual participant training data with a logarithmic function (see Supplementary Information: *Multi-session training study: Behavioral data analysis*).

## Multi-session training study: Imaging data acquisition and analysis

As reported in our previous study[17], all imaging data were collected at Wolfson Brain Imaging Centre, Cambridge UK, on a Siemens 3-Tesla Prisma (Siemens, Erlangen) with a 32-channel head coil. For detailed acquisition parameters see Supplementary Information: *Multi-session training study: MRI acquisition*. We used resting-state data and processed MPM maps from our previous study[17]. We generated the microstructural profile covariance (MPC) matrix for cortical myelination densities and functional connectivity (FC) similarity matrix for resting-state data. We then used MPC and FC to build microstructural and functional gradients (Fig. 1B) and network dispersion (Fig. 1F, Fig. S1). For detailed analysis pipeline and parameters, see Supplementary Information: *Multi-session training study: MRI data analysis*.

## Multi-session training study: Gene expression analysis

We processed microarray gene expression data from Allen Human Brain Atlas (AHBA)[10] with abagen toolbox (https://github.com/rmarkello/abagen)[53]. We followed a previous study[54] and extracted gene expressions for 16651 genes and parcellated them to the Schäefer-200 atlas (more details see Supplementary Information: *Microarray gene expression from Allen Human Brain Atlas*). Due to the limited availability of data for the right hemisphere (2 out of 6 individuals), we focused on the left hemisphere, where gene expression data are available for all six donors[55,56].

**PLS regression**. To identify genes that contribute to brain structure and function global changes, we followed the analysis pipeline from a previous study investigating gene expression and enriched pathways that contribute to cortical thickness changes in autism[57]. Partial least squares (PLS) has been previously used in neuroimaging studies with multi-collinear predictors or high data dimensionality[57]. We adopted PLS as the number of predictors (number of genes = 16651) exceeds the number of cortical nodes (number of nodes = 200[58]). In particular, we conducted a partial least squares regression (PLSr), a data reduction and regression technique related to principal component analysis (PCA) and ordinary least squares (OLS) regression (Fig. 1C),

$$X = TP^T + E$$

$$Y = UQ^T + F$$

Where

- $X$ is an $n \times m$ matrix of predictors, i.e. gene expression derived from AHBA
- $Y$ is an $n \times p$ matrix of responses, i.e. MPC principal gradients (MPC G1 – MPC G3) and FC principal gradients (FC G1 – FC G3) as response variables
- $T$ and $U$ are $n \times l$ matrices that are, respectively, projections of $X$ (the $X$ score, component or factor matrix) and projections of $Y$ (the $Y$ scores)

- $P$ and $Q$ are, respectively, $m \times l$ and $p \times l$ loading matrices
- and matrices $E$ and $F$ are the error terms, assumed to be independent and identically distributed random normal variables.

We standardized (z-scored) predictors and response variables before including them into the PLS model. To test the significance of the model, we permutated the response variables 10,000 times and performed a PLS regression for each permutation to generate a null distribution from our data[59]. Next, we assessed the stability of the predictor loadings to determine the significant predictors of the response variables[59] by generating 1,000 bootstrap samples from our data by sampling with replacement and performing a PLS regression for each bootstrap sample to generate a distribution per weight. To generate these distributions, we corrected the estimated components for axis rotation and reflection across bootstrap samples using Procrustes rotation and normalized the weights of the observed sample (that is, original data) to the standard deviation of the bootstrapped weights, resulting in z-score-like weights. FDR-adjusted with an FDR inverse quantile transformation correction was performed to account for winners curse bias[21] (R package FIQT). Genes that passed FDR correction of $p < 0.05$ ($|z| > 1.96$) were included in the enrichment analysis.

**Genetic enrichment analyses**. As for gene enrichment analysis, we used the Database for Annotation, Visualization and Integrated Discovery (DAVID)[20,60] (https://david.ncifcrf.gov/). The DAVID Knowledgebase built upon the DAVID Gene concept allows us to test for enriched brain functional-related gene groups, taking a list of gene IDs and annotating genes in a biological context, to test for significant PLSr-derived genes for each component with 16651 genes derived from AHBA as background genes. We performed enrichment test within 'homo sapiens' species and against tissue-type specific (GNF U133A quartile 79 tissue types) and reported significant results after Benjamini-Hochberg FDR correction (p < 0.05).

## tDCS and training intervention study: Experimental Design and Procedures

**Stimuli**. Stimuli presented were the same as for the multi-session training study (see Supplementary Information: *Stimuli*).

Experimental Design. All participants took part in a single brain imaging session during which they were randomly assigned to the Anodal or Sham group. Participants in the Anodal group received anodal tDCS on the right OCT, whereas participants in the Sham group did not receive stimulation. We collected whole-brain multi-parameter mapping (MPM) and rs-fMRI data before and after training while participants fixated on a cross at the center of the screen.

During training, participants were presented with Glass patterns and were asked to judge and indicate by button press whether the presented stimulus in each trial was radial or concentric. Two stimulus conditions (radial vs. concentric Glass patterns; 100 trials per condition), were presented for each training block. For each trial, a stimulus was presented for 300 ms and was followed by fixation (i.e. blank screen with a central fixation dot) while waiting for the participant's response (self-paced trials). Trial-by-trial feedback was provided by means of a visual cue (green tick for correct, red 'x' for incorrect) followed by a fixation dot for 500 ms before the onset of the next trial. Participants completed 9 blocks of 200 trials each. tDCS stimulation (Anodal or Sham) lasted 20 min, beginning at the start of block 3 and ending before block 6, ensuring the same amount of stimulation was applied during training across participants. Total duration of training did not vary between groups (Anodal group mean/stdev = 61.5/4.1 min, Sham group mean/stdev = 62.6/3.9 min, $T(39) = 0.81$, $p = 0.423$).

## tDCS and training intervention study: Imaging data acquisition

We collected MRI data on a 3T Siemens PRISMA scanner (Cognition and Brain Sciences Unit, Cambridge) using a 64-channel head coil. T1-weighted structural data (TR = 19.17 ms; TE = 2.31 ms; number of slices = 176; voxel size = 1 mm isotropic). We collected whole-brain multi-parameter mapping

(MPM) and rs-fMRI data using similar protocols as for the multi-session training study. For detailed acquisition parameters see Supplementary Information: *tDCS and training intervention study: MRI acquisition*).

### tDCS and training intervention study: Data analysis

**Behavior.** We measured behavioral performance per training block as the mean accuracy per 200 trials. To quantify learning-dependent changes in behavior, we computed the behavioral performance before and after stimulation as the average performance of blocks 1–2 (Pre-) and 6–9 (Post-stimulation), respectively.

**Image analysis.** We followed the same pipeline as for the multi-session training study. For detailed analysis pipeline see Supplementary Information: *tDCS and training intervention study: MRI data analysis*.

**Statistics and Reproducibility.** Repeated measures ANOVAs, multiple regressions, and T tests were performed in SPSS v19. We used Bonferroni correction for post-hoc comparison. Correction for gene expression analysis was performed with Benjamini–Hochberg FDR method[61]. For the tDCS intervention study, we normalized measurements from the Anodal to the Sham group by subtracting the mean value of the Sham group from the Anodal group for each measurement separately[26]. To ensure that our results were not simply due methodological choices, we tested: (a) different correction methods (i.e. Benjamini, Fisher Exact, FDR, and Bonferroni) for the gene expression enrichment analysis (Table S5), (b) three different Schäefer atlases (Supplementary Information: Sensitivity analyses; Fig. S2, Table S6). Further, we conducted leave-one-out cross-validation on the regression analyses for a) FC dispersion (Table S7), b) MPC dispersion predicting learning rate (Table S8).

### Reporting summary

Further information on research design is available in the Nature Portfolio Reporting Summary linked to this article.

## Data availability

Data are available in the Cambridge University repository (https://doi.org/10.17863/CAM.117105)[62]. AHBA microarray expression data can be downloaded via *abagen* toolbox[53,63].

## Code availability

Code is available in the Cambridge University repository (https://doi.org/10.17863/CAM.117105)[62].

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

## Acknowledgements

This work was supported by grants to ZK from the Biotechnology and Biological Sciences Research Council (H012508, BB/P021255/1), the Wellcome Trust (205067/Z/16/Z; 223131/Z/21/Z), to Z.K. and B.B. from the Douglas Avrith University of Cambridge/Montreal Neurological Institute-Hospital Neuroscience Collaboration. For the purpose of open access, the author has applied for a CC BY public copyright license to any Author Accepted Manuscript version arising from this submission. We would like to thank the MR physics and radiographer teams at the Wolfson Brian Imaging Centre and the Cognition and Brain Sciences Unit for their support with data collection, Matthew Davis and Benedikt Zoefel for their guidance on setting up tDCS in the scanner, Martina Callaghan, Nikolaus Weiskopf and the Wellcome Centre for Human Neuroimaging for providing for support with the MPM sequences and analyses, and Lilia Kukovska and Vicki Hodgson for help with data collection. We would like to thank Dr. Austin Benn and Richard Dear for their valuable insights on genetic analysis and gradient analysis.

## Author contributions

Conceptualization: L.Y.L., J.J.Z., P.F., V.M.K., B.B., V.W., R.A.I.B., Z.K; Data curation: L.Y.L., J.J.Z., P.F., V.M.K., Y.W., V.W., R.A.I.B., Z.K.; Formal Analysis: L.Y.L., J.J.Z., P.F., V.M.K., Y.W., V.W., R.A.I.B., Z.K.; Funding acquisition: B.B., Z.K.; Investigation: L.Y.L., J.J.Z., P.F., V.M.K.; Methodology: L.Y.L., J.J.Z., Y.W., V.W., R.A.I.B.; Project administration: L.Y.L., Z.K.; Resources: L.Y.L., Z.K., B.B., V.W., R.A.I.B.; Software: L.Y.L., J.J.Z., Y.W., V.W., R.A.I.B.; Supervision: Z.K., B.B., V.W., R.A.I.B.; Visualization: L.Y.L., Y.W., R.A.I.B.; Writing—original draft: L.Y.L., J.J.Z., P.F., V.M.K., B.B., V.W., R.A.I.B., Z.K.; Writing—review & editing: L.Y.L., J.J.Z., P.F., V.M.K., B.B., V.W., R.A.I.B., Z.K.

## Competing interests

The authors declare no competing interests.
