## [Transparent Peer Review file · Communications Biology]

Neurogenetic phenotypes of learning-dependent plasticity for improved perceptual decisions

Corresponding Author: Professor Zoe kourtzi

Version 0:

Reviewer comments:

Reviewer #1

(Remarks to the Author)

This study investigates the roles of gene expression and brain plasticity in adult perceptual learning. Using a visual discrimination task with patterns embedded in noise, the authors examined the relationships among gene expression, microstructural changes, and functional connectivity through multimodal imaging techniques, including quantitative MRI and resting-state fMRI. The behavioral training resulted in significant improvements in task performance, indicating that greater microstructural coherence between visual and fronto-parietal networks, along with enhanced functional segregation within the fronto-parietal network, was associated with faster learning rates. The findings revealed a correlation between gene expression in the occipital and prefrontal regions and learning-related changes in brain organization. Additionally, the application of anodal transcranial direct current stimulation (tDCS) during training improved performance by modifying functional connectivity rather than microstructural changes. This suggests that tDCS facilitates learning by increasing functional segregation in the frontal cortex while promoting coherence across networks. The authors conclude that the study highlights neurogenetic factors contributing to plasticity in perceptual decision-making networks, offering valuable insights into the interplay between genetic expression and the broader mechanisms of structural and functional plasticity involved in learning and adaptive behavior.

I am fascinated by the diverse range of behavioral, imaging, gene expression analysis, and brain stimulation technologies employed in this study. The behavioral, imaging, and brain stimulation experiments and analyses are conducted with high rigor. Although I have limited knowledge of gene expression analysis, I am eager to learn about its connection to perceptual learning. I have a few minor comments for the authors to consider:

1. The use of gradient analysis of MT and functional connectivity signals to characterize principal gradients of cortical microstructure (i.e., grey matter myelin) and functional connectivity beyond local processing is new to me. Could the authors provide additional rationale for this approach?
2. Is it possible to estimate gene expression for each individual based on their microstructural characteristics and relate these to individual differences in the behavioral task (e.g., learning rate, performance level, etc.)?
3. It would be clearer to break the following sentence into two: "We found that the first three PLS components explained a significant amount of variance ($p = 0.036, 0.0081, 0.038$; 5.32%, 5.68%, and 3.87% respectively, 10,000 permutations) in learning-dependent changes in cortical organization."

Reviewer #2

(Remarks to the Author)

This is a very interesting paper exploring a correlation between gene expression and modifications in visual and fronto-parietal networks induced by perceptual training. Subjects were trained to identify radial vs concentric glass patterns embedded in noise and demonstrated behavioral improvement over several sessions. The authors used the 200 ROI Schaefer parcellation dividing the cortical surface in equivolumetric regions and measured magnetization transfer saturation perpendicular to the flattened cortical sheet as well as functional connectivity coefficients at resting state between those regions. They then computed microstructure profile covariance (MPC) and functional connectivity similarity matrix principal gradients on the cortical surface and used Partial Least Square regression to see if/how gene expression profiles from the Allen Human Brain Atlas (corresponding to the 200 Schaefer parcellation) relate to pre- vs post- training changes seen in the spatial variation of microstructure and functional connectivity.

PLS weights for each gene were z-transformed and FDR-adjusted. The authors found that the first 3 independent gene

profile components explained a significant amount of variance (3.9-5.7%) in learning-dependent changes of cortical organization. The authors suggest that specific profiles of gene expression in occipital and prefrontal regions are associated with the learning-dependent changes that they see in microstructure organization and functional connectivity of visual and fronto-parietal networks. Specifically, FPN within network FC dispersion was positively correlated with learning rate (FPN-VN or within VN was not significant). FPN-VN between network MPC dispersion was negatively correlated with learning rate (within FPN, or within VN was not sig). For performance accuracy following "long-term" training, the FC dispersion did not matter, while VN within network MPC dispersion was positively correlated and VN-FPN MPC dispersion was negatively correlated with behavioral improvement (performance accuracy). In sum, higher structural coherence between visual and fronto-parietal networks, microstructural segregation within the visual network and functional segregation of regions within the fronto-parietal network relate to faster learning during multi-session training.

They also applied anodal tDCS in the visual cortex during training within a single session, and found that it boosts learning and is associated with altered functional connectivity between visual and fronto-parietal networks rather than with changing microstructural organization. Overall the results suggest a link between gene expression and plasticity in perceptual decision networks; functional plasticity at early stages of learning, and both functional and microstructural plasticity at later stages of learning (i.e. following longer-term training) in the adult human brain. They suggest that functional connectivity reflects flexible functional plasticity at early stages of learning, while slower micro-structural plasticity that may support learning consolidation occurs after longer-term training.

I have the following comments:

1) The results are correlative, so the claim that "tDCS stimulation in visual cortex during training boosts learning by altering functional connectivity" is not strongly supported by the experiments performed. This type of claim also recurs in other places in the manuscript. For example, it is possible that other factors such as better ability to focus attention on the task might play a role in the patterns seen and may not be directly related to perceptual learning per se. A more nuanced discussion of this possibility and a more careful exposition of the related arguments would help keep the possibilities clear in the mind of the reader.

2) the paper would benefit with a more detailed explanation of the techniques used in the main manuscript, including formulas where appropriate. The supplementary information explains in general the process referring to various packages for processing, but it would still be informative to outline the computations performed in detail via mathematical formulas. Along the same lines, the Figure 1 schematic could benefit from additional details, such as example data or visuals showing what the gradients that were correlated to the learning look like, to help readers acquire a more intuitive feel for the results.

3) It would be great if the individual participant data points could be shown in figures 4 and 2b to demonstrate spread of responses. In figure 4 connecting the individual subject points for pre- and post- groups would also allow the reader to view quickly each subjects responses during training.

4) The behavioral improvement effect in figure 4 is from 65 to 67 %, a relatively small difference. What differences shown in this figure are significant and to what level? -- How do the authors explain the pre-training difference between the sham and the anodal group?

5) A null distribution could be prepared by splitting the subjects at the baseline and pre-training time points into multiple randomly chosen group pairs having equal number of subjects. These group pairs can then undergo the same analysis and the question asked how many times the observed relationships pre- vs post- training would be reproduced by chance in this null set of pairs (for which there is no long term training). This empiric statistical approach would be reassuring in making certain the observations are related to training and not to some other factor that might not have been necessarily controlled by the statistical tests currently employed.

6) It would help to perform sensitivity analysis to ensure the parameters chosen during the data processing do not significantly modify the results. For example, are the results consistent, if different Schaffer parcellations are used? Does the choice of thresholding for gene expression significance alter the genetic associations observed?

7) It would be great, if possible to prepare an illustration of the differences observed in the structural and functional dispersion changes that occur with learning. The paper would benefit from a more nuanced discussion/explanation of what the employed measures of dispersion metrics mean in the context of the functional cortical networks. Is the connectivity between specific parcellation sub-regions account for the increased dispersion? A visual schematic/representation of the degree of dispersion changes across the nodes, demonstrating regions that manifest higher coherence or segregation and associated gene changes would be helpful.

8) in the supplementary information the authors say that changes before training (pre-training vs. baseline) in the FC dispersion predicted learning rate though with a negative relationship to learning rate. If I understand correctly, pre-training was the same type of session as baseline, except presented for the second time 5 days later. Why would the FC dispersion change be negatively related to learning "pre-training", while post-training (more repetitions of the same type of session, if I am not mistaken) it becomes positive? It would be great to have the authors comment on that

Minor:

1) I am a bit confused about the duration of the stimulation since the paradigm was self paced. If direct current stimulation was during training for sessions 3-5 how was it exactly 20 min for each subject ?

2) "Figure 3C shows that increased microstructure dispersion in VN relates to higher behavioral improvement, suggesting that higher segregation in structural organization across regions in the visual network after training relates to learning. In contrast, decreased microstructure dispersion between VN and FPN (Figure 3D) relates to higher behavioral improvement, suggesting that higher coherence in structural organization across networks after training relates to learning." -- Exchange figure panel references 3D and 3C ?

Reviewer #3

(Remarks to the Author)

Lee and colleagues combined multimodal neuroimaging with gene expression to identify transcriptomic correlates of the changes in perceptual decision abilities after training. They further used anodal tDCS on the occipital lobe and showed its impact on functional networks. However, this work has a very limited sample size and the manuscript is not well organized. I also have several major concerns about the current methodology. More analyses are absolutely necessary to support the current conclusion.

1. PLS correlation analysis was performed to examine gene transcriptions that are associated with changes in brain gradients. However, as far as I see, the authors did not consider the influence of spatial autocorrelation.

As well discussed in previous works, gene expressions of neighboring samples are tightly correlated. Therefore, when the spatial autocorrelation is not controlled, the downstream gene-set analysis will result in many false positives. Therefore, I would suggest the authors add another permutation analysis with brain maps/expression maps randomly permuted with spatial distance conserved, to avoid false positives in the analysis.

Fulcher, B.D., Arnatkeviciute, A. & Fornito, A. Overcoming false-positive gene-category enrichment in the analysis of spatially resolved transcriptomic brain atlas data. *Nat Commun* 12, 2669 (2021). <https://doi.org/10.1038/s41467-021-22862-1>

2. The authors argued that the observed genes are preferentially expressed in the prefrontal and occipital lobes. However, in SI table 2, most of the terms are not related to the brain. Therefore it's not fair to draw the conclusion that the identified genes are specifically expressed in prefrontal and occipital regions, not in other regions in the parietal or temporal lobe.

3. Similarly, it is also well known that brain gene expression data have several principal components, one of which is overexpressed in the occipital lobe. Could the author add any other analysis to show the specificity of the observed gene set?

Dear, R., Wagstyl, K., Seidlitz, J. et al. Cortical gene expression architecture links healthy neurodevelopment to the imaging, transcriptomics and genetics of autism and schizophrenia. *Nat Neurosci* 27, 1075–1086 (2024). <https://doi.org/10.1038/s41593-024-01624-4>

4. This work examined associations between brain network segregation and behavioral performance. The sample size is very small with $df = 15$ in the multiple regression analysis. I do not think this small sample size could make any robust statistical inference.

5. I would suggest the authors expand the current figure 1. The experiment design of the study is way more complicated than a simple transcriptome-imaging association analysis, including training sessions and tDCS stimulation sessions. A general overview of the designed experiments, analyses, and data samples would be beneficial.

6. The authors argued that post- vs pre- training changes in MPC and FC gradients were shown in Figure 1B. However, it's unclear what is shown in Figure 1B, is that t-value or effect size? Also, the color map isn't centered at zero, right? (I couldn't see that clearly as the resolution is a bit low). If so this will be misleading because colors in different brainmaps might indicate different effect sizes. I would also suggest the authors include the averaged gradients for each group separately.

Reviewer #4

(Remarks to the Author)

Version 1:

Reviewer comments:

Reviewer #1

(Remarks to the Author)

The authors have addressed all my concerns in this revision.

Reviewer #2

(Remarks to the Author)

Thank you for the detailed response.

Reviewer #3

(Remarks to the Author)

The authors have addressed my concerns. I do not have any further comment.

Reviewer #4

(Remarks to the Author)

This revised manuscript presents a well-executed and interdisciplinary study examining the neurogenetic underpinnings of learning-related plasticity. The combination of behavioral training, multimodal neuroimaging, transcriptomic data, and non-invasive brain stimulation is both ambitious and timely. The central idea—that gene expression patterns in occipital and prefrontal regions relate to structural and functional changes following perceptual learning—is conceptually interesting and explored through a thoughtful analytic framework.

The application of gradient analysis to link microstructural and functional brain organization is particularly effective, and the authors now explain its rationale and implementation more clearly. Their use of dispersion metrics to track network-level changes adds nuance to the interpretation of learning-dependent brain plasticity. While the results remain partly exploratory—especially given the modest sample size—the authors are transparent about these limitations and frame their conclusions appropriately.

The revisions address prior concerns in a thorough and convincing manner. Additional control analyses, including those accounting for spatial autocorrelation, strengthen the validity of the transcriptomic associations. The manuscript is now clearer and more detailed, especially in its methods and supplementary information, making it easier for other researchers to follow and potentially reproduce the work.

The study brings together diverse approaches to tackle an important question about how learning shapes the adult brain. I expect it will be of interest to researchers in systems neuroscience, cognitive neurogenetics, and learning. I support publication of the revised manuscript

Response to reviewers

Reviewer #1

1. The use of gradient analysis of MT and functional connectivity signals to characterize principal gradients of cortical microstructure (i.e., grey matter myelin) and functional connectivity beyond local processing is new to me. Could the authors provide additional rationale for this approach?

Thank you for the suggestion. We have revised the Introduction to clarify the rationale for using the gradient analysis approach to investigate the link between structural and functional plasticity.

Specifically, conventional analysis methods typically emphasize "local" measures of connections or comparisons. For example, for functional connectivity analysis, the most common approach calculates the FC matrix by computing pairwise Pearson correlation coefficients. This is considered a "local" measure, as it focuses on pairwise relationships between nodes.

In contrast, gradient analysis extends connectivity analysis from "local" to "global" by extracting the principal components (i.e. principal gradients) of functional connectivity or microstructure similarity matrix across the entire cortex. These principal gradients capture overarching spatial trends, showing how nodes are globally similar to each other across the whole brain, capturing network function within and across cortical networks.

Previous work has demonstrated that gradient analysis offers a spatial framework for organizing multiple large-scale networks. Further, it characterizes a spectrum of functional activity, ranging from unimodal to heteromodal regions, as seen in functional meta-analyses (Margulies et al., 2016; Smallwood et al., 2021; Laird et al., 2009) and microstructural gradients analyses (Paquola et al., 2019; Royer et al., 2021).

Importantly, gradient analysis allows us to link and compare microstructure and functional plasticity in the same information space; that is, capturing similarity (e.g. dispersion) in spatial gradients within and across cortical networks as markers of structural and functional brain plasticity.

The revised text writes:

'Previous work has demonstrated that gradient analysis offers a spatial framework for organizing multiple large-scale networks. Further, it characterizes a spectrum of functional activity, ranging from unimodal to heteromodal regions, as demonstrated by functional meta-analyses¹² and microstructural gradients analyses¹³. Finally, gradient analysis allows us to link and compare microstructure and functional plasticity in the same information space by extracting markers of structural and functional brain plasticity based on the similarity spatial gradients within and across cortical networks.'

2. Is it possible to estimate gene expression for each individual based on their microstructural characteristics and relate these to individual differences in the behavioral task (e.g., learning rate, performance level, etc.)?

Thank you for this interesting suggestion. There are existing methods for estimating gene expression at the individual level using phenotype data. One widely used method is the transcriptome-wide association study (TWAS; Mai et al., 2023), which identifies relationships between gene expression and specific human traits. TWAS typically follows a three-step process:

- (a) A genetic regulation model is trained using an independent reference panel that includes both genetic data and measured gene expression.
- (b) The trained model, incorporating regulatory weights, is then used to impute gene expression for individuals in large-scale genome-wide association studies (GWAS).
- (c) Finally, associations between predicted gene expression and traits of interest are examined to determine potential gene-trait regulatory relationships.

However, applying TWAS to our study is challenging for the following reasons. First, we lack a reference dataset with individual genetic profiles and gene expression data that is needed for this approach. Our gene-expression levels originate from the Allen Human Brain Atlas (AHBA). AHBA provides gene expression data from four individuals (and six samples) that is not adequate for constructing and training a genetic regulation model. Further AHBA does not provide any data related to the phenotype relevant for our study (i.e. training on a visual discrimination task).

Second, TWAS studies often leverage large population-specific datasets to enhance statistical power. For example, Barbeira et al. (2023) utilised genetic data from 250,000 individuals to identify genes associated with height, while Valette et al. (2021) analysed data from 56,167 asthma cases and 352,255 controls to pinpoint asthma-related loci. Our study focuses on linking genetics to learning-dependent changes in functional and microstructure connectivity due to training on a visual discrimination task. To the best of our knowledge, there are no existing large-scale GWAS data on this kind of perceptual learning task and our current smaller-scale study is underpowered for this type of TWAS analysis.

Despite these limitations, exploring gene-trait associations remains a promising direction for future research. Larger datasets allow us to investigate how individual variations in gene expression—potentially inferred from neuroimaging-derived microstructural characteristics—relate to behaviour. A complementary approach may involve linking imaging-derived phenotypes with gene expression patterns from AHBA using spatial transcriptomics frameworks, allowing indirect inference of gene-behaviour relationships.

In the revised manuscript, we discuss the limitations of our study and the potential of this approach for future work. In particular, the text writes:

'Future work is needed, considering a broader range of individual variability and using methods for estimating gene expression at the individual level (e.g. transcriptome-wide association study; TWAS³⁷), to identify relationships between gene expression and specific human traits. TWAS studies leverage large population-specific datasets to enhance statistical power. Although there are no existing large-scale GWAS data on perceptual learning and our study sample is not large enough for TWAS analysis, exploring gene-trait associations remains an interesting direction for future research.'

3. *It would be clearer to break the following sentence into two: "We found that the first three PLS components explained a significant amount of variance ($p = 0.036, 0.0081, 0.038$; 5.32%, 5.68%, and 3.87% respectively, 10,000 permutations) in learning-dependent changes in cortical organization.*

Thank you for pointing this out. We have revised the sentence to write:

'We found that the first three PLS components explained significantly variance in learning-dependent microstructure and functional changes (10,000 permutations test; PLS1: $p = 0.036$, PLS2: 0.0081, PLS3: 0.038). Specifically, these components accounted for 5.32% (PLS1), 5.68% (PLS2), 3.87% (PLS3) of the variance.'

References

Margulies DS, Ghosh SS, Goulas A, Falkiewicz M, Huntenburg JM, Langs G, Bezgin G, Eickhoff SB, Castellanos FX, Petrides M, Jefferies E, Smallwood J. Situating the default-mode network along a principal gradient of macroscale cortical organization. *Proc Natl Acad Sci U S A*. 2016 Nov 1;113(44):12574-12579. doi: 10.1073/pnas.1608282113. Epub 2016 Oct 18. PMID: 27791099; PMCID: PMC5098630.

Smallwood, J., Bernhardt, B.C., Leech, R. *et al.* The default mode network in cognition: a topographical perspective. *Nat Rev Neurosci* **22**, 503–513 (2021). <https://doi.org/10.1038/s41583-021-00474-4>

Laird AR, Eickhoff SB, Li K, Robin DA, Glahn DC, Fox PT. Investigating the functional heterogeneity of the default mode network using coordinate-based meta-analytic modeling. *J Neurosci*. 2009 Nov 18;29(46):14496-505. doi: 10.1523/JNEUROSCI.4004-09.2009.

Paquola C, Vos De Wael R, Wagstyl K, Bethlehem RAI, Hong S-J, Seidlitz J, et al. (2019) Microstructural and functional gradients are increasingly dissociated in transmodal cortices. *PLoS Biol* 17(5): e3000284. <https://doi.org/10.1371/journal.pbio.3000284>

Royer J, Paquola C, Larivière S, Vos de Wael R, Tavakol S, Lowe AJ, Benkarim O, Evans AC, Bzdok D, Smallwood J, Frauscher B, Bernhardt BC. Myeloarchitecture gradients in the human insula: Histological underpinnings and association to intrinsic functional connectivity. *Neuroimage*. 2020 Aug 1;216:116859. doi: 10.1016/j.neuroimage.2020.116859. Epub 2020 Apr 20. PMID: 32325211.

Mai, J., Lu, M., Gao, Q. et al. Transcriptome-wide association studies: recent advances in methods, applications and available databases. *Commun Biol* 6, 899 (2023). <https://doi.org/10.1038/s42003-023-05279-y>

Barbeira AN, Dickinson SP, Bonazzola R, Zheng J, Wheeler HE, Torres JM, Torstenson ES, Shah KP, Garcia T, Edwards TL, Stahl EA, Huckins LM; GTEx Consortium; Nicolae DL, Cox NJ, Im HK. Exploring the phenotypic consequences of tissue specific gene expression variation inferred from GWAS summary statistics. *Nat Commun*. 2018 May 8;9(1):1825. doi: 10.1038/s41467-018-03621-1. PMID: 29739930; PMCID: PMC5940825.

Valette, K. et al. Prioritization of candidate causal genes for asthma in susceptibility loci derived from UK Biobank. *Commun. Biol.* 4, 700 (2021). doi: 10.1038/s42003-021-02227-6.

Reviewer #2

1) The results are correlative, so the claim that "tDCS stimulation in visual cortex during training boosts learning by altering functional connectivity" is not strongly supported by the experiments performed. This type of claim also recurs in other places in the manuscript. For example, it is possible that other factors such as better ability to focus attention on the task might play a role in the patterns seen and may not be directly related to perceptual learning per se. A more nuanced discussion of this possibility and a more careful exposition of the related arguments would help keep the possibilities clear in the mind of the reader.

Thank you for the suggestion. We have revised the text throughout the manuscript to write:

'tDCS stimulation in visual cortex during training alters functional connectivity and improves task performance'.

We discuss this point further in the revised manuscript.

'Note that these changes were measured before vs. after the tDCS intervention when there was no tDCS stimulation or performance feedback provided to the participants. tDCS may enhance attention during the intervention. However, following the intervention task performance was improved suggesting that the task was easier and therefore less attentionally demanding in comparison to the Sham group, where the task remained demanding and may have required more attentional resources.'

2) the paper would benefit with a more detailed explanation of the techniques used in the main manuscript, including formulas where appropriate. The supplementary information explains in general the process referring to various packages for processing, but it would still be informative to outline the computations performed in detail via mathematical formulas. Along the same lines, the Figure 1 schematic could benefit from additional details, such as example data or visuals showing what the gradients that were correlated to the learning look like, to help readers acquire a more intuitive feel for the results.

Thank you for the suggestions. We have added more detailed explanation of the analysis pipeline in the: a) SI on equivolumetric surface construction technique (*Multi-session training study: MRI data analysis - Microstructural profile covariance (MPC) and microstructural gradients*), b) main text (*Methods - PLS regression*).

We have also revised Figure 1F and included Figure S1 to provide an illustration of the gradients before and after training.

3) It would be great if the individual participant data points could be shown in figures 4 and 2b to demonstrate spread of responses. In figure 4 connecting the individual subject points for pre- and post-groups would also allow the reader to view quickly each subjects responses during training.

Thank you for this suggestion. We have revised Figure 2b and Figure 4 accordingly.

4) The behavioral improvement effect in figure 4 is from 65 to 67 %, a relatively small difference. What differences shown in this figure are significant and to what level? -- How do the authors explain the pre-training difference between the sham and the anodal group?

Thank you for raising this point. In the revised text and Figure 4 caption we have clarified the statistics to provide more detail. In particular, we observed a significant interaction: Group (Anodal, Sham) x Block (Pre-, Post-intervention) ($F(1, 43) = 4.72, p = 0.035$; Figure 4A), but no significant main effect of Block ($F(1, 43) = 1.32, p = 0.26$) nor significant main effect of Group ($F(1, 43) = 2.98, p = 0.091$). Post-hoc comparison showed that participants in the Anodal group ($p = 0.019$, Bonferroni corrected) but not in the Sham group ($p = 0.49$) improved after training. The revised Figure 4 shows box plots and individual data for both groups. Differences in performance between groups before stimulation were not significant ($t(43)=1.02, p = 0.31$).

5) A null distribution could be prepared by splitting the subjects at the baseline and pre-training time points into multiple randomly chosen group pairs having equal number of subjects. These group pairs can then undergo the same analysis and the question asked how many times the observed relationships pre- vs post- training would be reproduced by chance in this null set of pairs (for which there is no long term training). This empiric statistical approach would be reassuring in making certain the observations are related to training and not to some other factor that might not have been necessarily controlled by the statistical tests currently employed.

Thank you for this suggestion. We have conducted the suggested permutation analysis; that is, we run the regression analyses 1000 times using random pairs of baseline vs. pre data for EV within network dispersion, FPN within network dispersion and EV-FPN between network dispersion. For FC dispersion, only 46 (out of 1000) models showed the relationship between differences pre- vs post-training FC and behaviour ($p = 0.954$). For MPC dispersion, only 45 (out of 1000) models showed the relationship between differences pre- vs post-training MPC and behavior ($p = 0.955$). These results strengthen the specificity of the learning-dependent changes before vs. after training, suggesting that any differences between baseline and pre-training are unlikely to contribute significantly to the learning effects we observed.

We have now included this analysis in the revised manuscript: Results - *Learning-dependent changes in occipito-frontal brain networks*.

6) It would help to perform sensitivity analysis to ensure the parameters chosen during the data processing do not significantly modify the results. For example, are the results consistent, if different Schaefer parcellations are used? Does the choice of thresholding for gene expression significance alter the genetic associations observed?

Thank you for this suggestion. To ensure that our results were not simply due methodological choices, we tested: a) three different Schaefer atlases, b) different correction methods for the gene expression enrichment analysis.

For Schaefer parcellation, we focused on the main predictors (i.e. FPN within-network FC dispersion, VN-FPN between-network MPC dispersion) of learning rate. We calculated dispersion using 3 different atlases (Schaefer 200, 300, 400) and generated correlation matrices across atlases. Our results show that FPN within-network FC dispersions and VN-FPN between-network MPC dispersions are highly correlated across different Schaefer atlases (Figure S1) and remain significant predictors of learning rate (Table S6). We have now included these results in *SI Sensitivity analysis - Schaefer parcellations*, Figure S2, and Table S6.

For gene expression extraction, we refer to the practical guide (Arnatkevičiūtė et al. 2019) and follow the Standardizing workflows (Markello et al. 2021). To test sensitivity, we investigated different correction methods for gene expression enrichment test. The results remained the same across different correction methods (i.e. Benjamini, Fisher Exact, FDR, and Bonferroni). We have now included these results in *SI Sensitivity analysis - Gene expression enrichment correction methods*, and Table S5.

7) It would be great, if possible to prepare an illustration of the differences observed in the structural and functional dispersion changes that occur with learning

Thank you for the suggestion. We have revised Figure 1F to demonstrate how FPN within-network FC dispersion changes with training.

The paper would benefit from a more nuanced discussion/explanation of what the employed measures of dispersion metrics mean in the context of the functional cortical networks. Is the connectivity between specific parcellation sub-regions account for the increased dispersion?

Further, in the revised text we have clarified better the dispersion metrics and the rationale for using gradient analysis and dispersion to investigate the link between structural and functional plasticity (see also Reviewer 1, point 1). Specifically, we calculated two types of dispersion: within-network dispersion and between-network dispersion. The connectivity *changes* between specific parcellation sub-regions in the same network (e.g. FPN) account for the increased within-network dispersion, and the connectivity *changes* between sub-regions in different networks (e.g. FPN and EV) account for the increased between-network dispersion. We have now explained more clearly in the revised manuscript how dispersion was calculated and what dispersion changes mean.

In particular, the text writes:

'We then estimated within- and between-network dispersion for MPC and FC principal gradients, a metric developed to quantify the variability in connectivity patterns and structural complexity across different brain regions. We defined the dispersion space by the values along the first three gradients (Figure 1F, Figure S1). Within network dispersion was quantified as sum squared Euclidean distance of network nodes to the network centroid at individual participant level. Between network dispersion was calculated as the Euclidean distance between network centroids. Dispersion indicates microstructural and functional network segregation¹⁶ (Figure 1F, Figure S1; Methods – Multi-session training study: Data analysis). That is, higher dispersion indicates higher segregation, while lower dispersion indicates higher coherence within or between networks.'

'Previous work has demonstrated that gradient analysis offers a spatial framework for organizing multiple large-scale networks. Further, it characterizes a spectrum of functional activity, ranging from unimodal to heteromodal regions, as demonstrated by functional meta-analyses¹² and microstructural gradients analyses¹³. Finally, gradient analysis allows us to link and compare microstructure and functional plasticity in the same information space by extracting markers of structural and functional brain plasticity based on the similarity spatial gradients within and across cortical networks.'

A visual schematic/representation of the degree of dispersion changes across the nodes, demonstrating regions that manifest higher coherence or segregation and associated gene changes would be helpful

We have revised Figure 1F to show how changes in FPN within-network FC dispersion relate to training. The shaded area for the post-training session (blue) is larger than that for the pre-training session (red), indicating that the nodes representing functional connectivity similarity within the FPN network are more spread-out after training (i.e. higher network segregation). We are not able to include gene changes as a) the gene expression data were derived from AHBA, b) we do not have gene expression data for participants in our study.

8) in the supplementary information the authors say that changes before training (pre-training vs. baseline) in the FC dispersion predicted learning rate though with a negative relationship to learning rate. If I understand correctly, pre-training was the same type of session as baseline, except presented for the second time 5 days later. Why would the FC dispersion change be negatively related to learning "pre-training", while post-training (more repetitions of the same type of session, if I am not mistaken) it becomes positive? It would be great to have the authors comment on that pre-training was the same type of session as baseline, except presented for the second time 5 days later.

Thank you for this question. We confirm that pre-training and baseline followed the same design. We have now clarified this in the revised SI - *Multi-session training study: Behavioral session design*.

Why would the FC dispersion change be negatively related to learning "pre-training", while post-training (more repetitions of the same type of session, if I am not mistaken) it becomes positive

Thank you for this question. Due to the relatively small sample size, we have replaced this analysis with a more robust permutation analysis.

Specifically, we run the regression analyses 1000 times using random pairs of baseline vs. pre data for EV within network dispersion, FPN within network dispersion and EV-FPN between network dispersion. For FC dispersion, only 46 (out of 1000) models showed the relationship between differences pre- vs post-training FC and behaviour ($p = 0.954$). For MPC dispersion, only 45 (out of 1000) models showed the relationship between differences pre- vs post-training MPC and behavior ($p = 0.955$). These results strengthen the specificity of the learning-dependent changes before vs after training, suggesting that any differences between baseline and pre-training are unlikely to contribute significantly to the learning effects we observed.

Minor:

1) *I am a bit confused about the duration of the stimulation since the paradigm was self paced. If direct current stimulation was during training for sessions 3-5 how was it exactly 20 min for each subject?*

Thank you for pointing this out. We apologise for the lack of detailed information. We have revised the text to clarify this point. In particular, the revised text writes:

‘Participants completed 9 blocks of 200 trials each. tDCS stimulation (anodal or sham) lasted 20 min, beginning at the start of block 3 and ending before block 6, ensuring the same amount of stimulation was applied during training across participants. Total duration of training did not vary between groups, Anodal group mean/stdev = 61.5/4.1 min, Sham group mean/stdev = 62.6/3.9 min, $T(39) = 0.81$, $p = 0.423$. ‘

2) *"Figure 3C shows that increased microstructure dispersion in VN relates to higher behavioral improvement, suggesting that higher segregation in structural organization across regions in the visual network after training relates to learning. In contrast, decreased microstructure dispersion between VN and FPN (Figure 3D) relates to higher behavioral improvement, suggesting that higher coherence in structural organization across networks after training relates to learning." -- Exchange figure panel references 3D and 3C ?*

Thank you for pointing this out, we apologize for the mistake. We have revised the figures including individual data and corrected the figure references.

Reference

Ziminski, J. J., Frangou, P., Karlaftis, V. M., Emir, U. & Kourtzi, Z. (2023) Microstructural and neurochemical plasticity mechanisms interact to enhance human perceptual decision-making. *PLoS Biol* 21, e3002029.

Arnatkeviciūtė A., Fulcher, B. D. , & Fornito, A.. (2019). A practical guide to linking brain-wide gene expression and neuroimaging data. *NeuroImage*, 189.

Markello RD, Arnatkeviciute A, Poline JB, Fulcher BD, Fornito A, Misic B. (2021) Standardizing workflows in imaging transcriptomics with the abagen toolbox. *Elife*. Nov 16;10:e72129. doi: 10.7554/eLife.72129. PMID: 34783653; PMCID: PMC8660024.

Reviewer #3:

1. PLS correlation analysis was performed to examine gene transcriptions that are associated with changes in brain gradients. However, as far as I see, the authors did not consider the influence of spatial autocorrelation.

As well discussed in previous works, gene expressions of neighboring samples are tightly correlated. Therefore, when the spatial autocorrelation is not controlled, the downstream gene-set analysis will result in many false positives. Therefore, I would suggest the authors add another permutation analysis with brain maps/expression maps randomly permuted with spatial distance conserved, to avoid false positives in the analysis.

Fulcher, B.D., Arnatkeviciute, A. & Fornito, A. Overcoming false-positive gene-category enrichment in the analysis of spatially resolved transcriptomic brain atlas data. *Nat Commun* 12, 2669 (2021). <https://doi.org/10.1038/s41467-021-22862-1>

Thank you for this suggestion. To address this issue and account for the spatial autocorrelations, we run two control analyses with: a) the raw gene expressions, b) denoised gene expressions (i.e. regressing out the 1st PCA component from the raw gene expression). In brief, we generated 1000 null brain models for gradients using the brainspace toolbox (Vos de Wael et al., 2020) and updated the permutation test with the new null brain gradients. We then followed the same analysis pipeline as reported in the manuscript (i.e., PLS with permutation test, bootstrapping, FDR correction and enrichment test). This analysis, after accounting for spatial autocorrelations showed the same result as previously reported; that is, gene expression preferentially in occipital and prefrontal regions is associated with learning-dependent changes in microstructure and functional connectivity.

We have included this analyses in the revised manuscript and provided additional tables in the SI (Table S3, Table S4).

2. The authors argued that the observed genes are preferentially expressed in the prefrontal and occipital lobes. However, in SI table 2, most of the terms are not related to the brain. Therefore it's not fair to draw the conclusion that the identified genes are specifically expressed in prefrontal and occipital regions, not in other regions in the parietal or temporal lobe.

Thank you for this question. We apologise for the confusion. We have now clarified in the revised text that 'This analysis identified significant enrichment for genes in PLS component 1 that are preferentially expressed within occipital ($p < 0.001$, Benjamin-Hochberg corrected) and prefrontal ($p = 0.032$, Benjamin-Hochberg corrected) lobes among brain tissues.' and revised Table S2.

In particular, we trained the PLS model using cortical gene expression data against cortical signal changes; that is, other tissues did not contribute to the selection of enriched genes. This approach allows us to interpret the results specifically in relation to brain regions.

To our knowledge, there are no existing tools that perform gene expression enrichment analysis exclusively for brain regions. Therefore, we conducted the enrichment test using the GNF U133A dataset, which includes 79 tissue types, covering major brain regions. In the study, we reported only the brain regions that survived multiple comparison correction. A complete table of adjusted p-values from the enrichment test is provided (Table S2), supporting the point that the observed genes are preferentially expressed in the prefrontal and occipital lobes.

3. Similarly, it is also well known that brain gene expression data have several principal components, one of which is overexpressed in the occipital lobe. Could the author add any other analysis to show the specificity of the observed gene set?

Dear, R., Wagstyl, K., Seidlitz, J. et al. Cortical gene expression architecture links healthy neurodevelopment to the imaging, transcriptomics and genetics of autism and schizophrenia. *Nat Neurosci* 27, 1075–1086 (2024). <https://doi.org/10.1038/s41593-024-01624-4>

Thank you for this suggestion. To test the specificity of our analysis, taking into account that first components of genetic expressions are overexpressed in the occipital lobe, we regressed out the 1st PCA component of the gene expression from the raw data and rerun the analysis. The results remain the same as previously reported; that is, gene expression preferentially in occipital and prefrontal regions is associated with learning-dependent structural organization and functional connectivity changes.

We have now included this analysis in the revised manuscript (Results - *Genetic signatures of learning-dependent plasticity in occipital and prefrontal regions*; Table S4)

4. This work examined associations between brain network segregation and behavioral performance. The sample size is very small with $df = 15$ in the multiple regression analysis. I do not think this small sample size could make any robust statistical inference.

Thank you for raising this point. To ensure robustness of our findings, despite the relatively small sample size, we conducted a permutation test ($n_{perm} = 1000$) that allows us to assess whether our results are statistically meaningful (i.e. related to training). This analysis (1,000 permutations) showed significant relationships between a) FC network dispersion ($p = 0.001$), b) MPC network dispersion ($p = 0.028$) and outcome (i.e. learning rate).

Further, we conducted leave-one-out cross-validation to assess the robustness of the regression analyses for a) FC dispersion (Table S7), b) MPC dispersion (Table S8) predicting learning rate. The results remain significant, suggesting higher functional segregation of regions within FPN network and higher coherence in structural organization between FPN and EV networks after training relates to learning.

We have now included this analysis in the revised manuscript and Supplementary Information (Table S7, S8).

5. I would suggest the authors expand the current figure 1. The experiment design of the study is way more complicated than a simple transcriptome-imaging association analysis, including training sessions and tDCs stimulation sessions. A general overview of the designed experiments, analyses, and data samples would be beneficial.

Thank you for this suggestion. Figure 1 aims to illustrate the link between genetic expression and gradient analyses. We have revised the figure to provide more detail on this. Further, following your suggestion, we have provided a general overview of the experiments in the Methods section. In particular, the revised text writes:

‘Overview

‘We conducted two studies: a) multi-session training study, b) intervention study. For both studies, participants were trained to discriminate radial vs. concentric Glass patterns embedded in noise. The multi-session training study investigated the link between gene expression, and learning-dependent functional and microstructural plasticity in perceptual decision networks. In this study, participants underwent multiple training sessions. MRI and behavioral test data were collected at baseline, pre- and post-training sessions. In the intervention study, we used anodal tDCS stimulation in visual cortex during scanning to compare learning-dependent network changes between the stimulation (Anodal tDCS during training) and the no stimulation (Sham tDCS during training) groups. This allowed us to test directly the link between microstructure and functional plasticity with behavioral improvement. For both studies, we collected resting-fMRI and MPM data before and after training. We generated the microstructural profile covariance (MPC) matrix for cortical myelination densities and functional connectivity (FC) similarity matrix for resting-state data. We then used MPC and FC to build microstructural and functional gradients and calculate network dispersion that reflects the network segregation and coherence.’

6. The authors argued that post- vs pre- training changes in MPC and FC gradients were shown in Figure 1B. However, it's unclear what is shown in Figure 1B, is that t-value or effect size? Also, the

color map isn't centered at zero, right? (I couldn't see that clearly as the resolution is a bit low). If so this will be misleading because colors in different brainmaps might indicate different effect sizes. I would also suggest the authors include the averaged gradients for each group separately.

We apologise for the confusion. Figure 1B shows the gradient value, rather than T-value or effect size. We have revised the Figure1B caption to clarify this. We have included an additional figure (Figure 1F) to show how gradient dispersion changes differ before vs after training.

Reference

Frangou, P., Correia, M. & Kourtzi, Z. GABA, not BOLD, reveals dissociable learning-dependent plasticity mechanisms in the human brain. *Elife* 7, (2018).

Vos de Wael R, Benkarim O, Paquola C, Lariviere S, Royer J, Tavakol S, Xu T, Hong S, Langs G, Valk S, Masic B, Milham M, Margulies D, Smallwood J, Bernhardt B (2020). BrainSpace: a toolbox for the analysis of macroscale gradients in neuroimaging and connectomics datasets. *Commun Biol* 3, 103.